# Glucose Detection of 4-Mercaptophenylboronic Acid-Immobilized Gold-Silver Core-Shell Assembled Silica Nanostructure by Surface Enhanced Raman Scattering

**DOI:** 10.3390/nano11040948

**Published:** 2021-04-08

**Authors:** Xuan-Hung Pham, Bomi Seong, Eunil Hahm, Kim-Hung Huynh, Yoon-Hee Kim, Jaehi Kim, Sang Hun Lee, Bong-Hyun Jun

**Affiliations:** 1Department of Bioscience and Biotechnology, Konkuk University, Seoul 143-701, Korea; phamricky@gmail.com (X.-H.P.); iambomi33@konkuk.ac.kr (B.S.); greenice@konkuk.ac.kr (E.H.); huynhkimhung82@gmail.com (K.-H.H.); hilite2201@naver.com (Y.-H.K.); susia45@gmail.com (J.K.); 2Department of Chemical and Biological Engineering, Hanbat National University, Daejeon 34158, Korea; sanghunlee@hanbat.ac.kr

**Keywords:** surface enhanced Raman scattering, gold-silver core-shell, gold-silver core-shell assembled silica nanostructure, hydrogen peroxide, glucose detection, 4-mercaptophenylboronic acid

## Abstract

The importance of glucose in many biological processes continues to garner increasing research interest in the design and development of efficient biotechnology for the sensitive and selective monitoring of glucose. Here we report on a surface-enhanced Raman scattering (SERS) detection of 4-mercaptophenyl boronic acid (4-MPBA)-immobilized gold-silver core-shell assembled silica nanostructure (SiO_2_@Au@Ag@4-MPBA) for quantitative, selective detection of glucose in physiologically relevant concentration. This work confirmed that 4-MPBA converted to 4-mercaptophenol (4-MPhOH) in the presence of H_2_O_2_. In addition, a calibration curve for H_2_O_2_ detection of 0.3 µg/mL was successfully detected in the range of 1.0 to 1000 µg/mL. Moreover, the SiO_2_@Au@Ag@4-MPBA for glucose detection was developed in the presence of glucose oxidase (GOx) at the optimized condition of 100 µg/mL GOx with 1-h incubation time using 20 µg/mL SiO_2_@Au@Ag@4-MPBA and measuring Raman signal at 67 µg/mL SiO_2_@Au@Ag. At the optimized condition, the calibration curve in the range of 0.5 to 8.0 mM was successfully developed with an LOD of 0.15 mM. Based on those strategies, the SERS detection of glucose can be achieved in the physiologically relevant concentration range and opened a great promise to develop a SERS-based biosensor for a variety of biomedicine applications.

## 1. Introduction

Today, diabetes mellitus is a global problem issue with approximately 200 million people worldwide. It is of the principal causes of morbidity and disability and is largely responsible for kidney failure, heart disease, and so on [1]. A monitoring of blood glucose is of important for diabetes care [2]. Therefore, the development of a simple, rapid, and accurate method for glucose detection is highly essential, because blood glucose level is closely associated with diabetes, and hypoglycemia [3,4]. Various methods for glucose detection have been reported, including high-performance liquid chromatography (HPLC) [5], colorimetric detection [6,7,8,9,10], electrochemical detection [11,12,13], fluorescence detection [14,15,16], and surface enhanced Raman scattering (SERS) [17,18,19,20,21,22,23]. Compared with HPLC, SERS exhibited significant advantages, such as rapid, non-destructive, molecular fingerprinting, ultrasensitive, and selectivity [24,25,26,27]. Therefore, various nanoparticles (NPs) such as silver and gold nanostructures have been developed as SERS substrates for glucose detection [17,18,19,20,21].

However, a low affinity with metal surface and Raman scattering cross-section of polarizability limited the detection of glucose by SERS [17,22,28,29]. To achieve a high affinity and selectivity for glucose, a various boronic acid-based Raman reporter have been used to capture glucose selectively on the substrate [17,19,28,29,30,31,32,33]. Most of those studies mainly focused on the specific binding of glucose with the boronic acid motif in 4-MPBA, leading to a significant increase in the absolute intensity of the SERS signal of 4-MPBA, which was ascribed to the orientation change and the charge transfer effect [17,18,19,21,34]. An active SERS substrate based on Ag@4-MPBA nanorod was reported for quantitative detection of glucose [17]. However, the Ag nanorods preparation can sometimes be complicated. On the other hand, the self-condensation of 4-MPBA in order to form anhydride due to the association of OH− with 4-MPBA in various pH media affects the binding of glucose through the formation of esters, which may lead to the properties of the functional molecules being different from those in a solution [18]. Furthermore, the calculation of the peak area of SERS band was so complicated, resulting in a narrow dynamic linear range of glucose concentration. Furthermore, the composite of Au nanoparticles and Porphyrin-Based Metal-Organic Framework for SERS was used to detect glucose with high selectivity and sensitivity [35]. However, the Porphyrin-Based Metal-Organic Framework (Cu-TCPP(Fe) MOF using surfactant caused the binding of surfactant on the surface of MOF nanosheet and blocked some the actives sites of MOFs [36]. It results in the difficulty in synthesizing well-dispersed 2D-MOF nanosheets without blocking their active sites.

Recently, our group developed a gold-silver core-shell assembled silica nanostructure (SiO_2_@Au@Ag) based on the Au-seed-mediated growth of Ag on the surface of a SiO_2_ NP template. The presence of Au seeds facilitates and allows for the precise control of the distance and uniform of the Ag shell on SiO_2_ NPs as well as tuning the optical property of SiO_2_@Au@Ag in the visible to near IR region [37,38,39]. The presence of the silica template facilitates the density, size, and shape, and the nanogap of Au-Au on the SiO_2_ surface was easily controlled by SiO_2_ core, and generating a homogenous SERS substrate of SiO_2_@Au@Ag [39]. As a result, a high-strength and reliable SiO_2_@Au@Ag based-SERS substrates with high reproducibility and 4.2 × 10^6^ fold-enhancement as compared to without the nanomaterial was developed for the enzyme-linked immunosorbent assay (ELISA) [40,41]. Simultaneously, a Raman reporter was also immobilized between the Au and Ag layer to play the role of an internal standard for the accurate detection of the pesticide thiram [42,43,44]. However, their use in SERS detection for glucose requires further investigation. In this study, a sensitive and selective analytical method using the basic structure of SiO_2_@Au@Ag NPs incubated in 4-MPBA and enzyme glucose oxidase was developed for glucose detection using SERS. The evaluation of the SERS band toward glucose level was calculated by the ratio of SERS intensity of intrinsic internal standard 4-MPBA, which was simpler than that based on the peak area of SERS bands.

## 2. Materials and Methods

### 2.1. Chemicals and Reagents

Tetraethylorthosilicate (TEOS), 3-aminopropyltriethoxysilane (APTS), silver nitrate (AgNO_3_), chloroauric acid (HAuCl_4_), tetrakis(hydroxymethyl)phosphonium chloride (THPC), ascorbic acid (AA), polyvinylpyrrolidone (PVP), phosphate buffer saline (PBS), Tween 20, 4-mercaptophenylboronic acid (4-MPBA), 4-mercaptophenol (4-MPhOH), 4-methylbenzenethiol (4-MBT), thiophenol, 4-glucose, hydrogen peroxide (H_2_O_2_), and glucose oxidase (GOx) were purchased from Sigma–Aldrich (USA). Ethanol (EtOH) and aqueous ammonium hydroxide (NH_4_OH, 27%) were purchased from Daejung (South Korea); and ultrapure water (18.2 MΩ cm) was produced by a Millipore water purification system (EXL water purification; Vivagen Co., Seongnam, Gyeonggi-do, Korea).

### 2.2. Preparation of SiO_2_@Au@Ag

The SiO_2_@Au@Ag NPs were prepared in accordance with the steps outlined in [38]. The SiO_2_@Au NPs were prepared by incubating 10 mL of Au NP suspension with 2 mL of aminated silica NPs overnight. The colloids were centrifuged and washed thoroughly using EtOH. The NPs were then re-dispersed in 2.0 mL of complete EtOH to obtain 1 mg/mL SiO_2_@Au NPs in EtOH.

The SiO_2_@Au@Ag NPs were prepared in an aqueous medium via the reduction and deposition of Ag using ascorbic acid onto SiO_2_@Au NPs in PVP. Moreover, 200 µL of SiO_2_@Au (1000 µg/µL) was briefly dispersed in 9.8 mL of water that contained 10 mg of PVP, which was kept still for 30 min. Thereafter, 20 µL of silver nitrate (10 mM) was added to the suspension, followed by the addition of 40 µL of ascorbic acid (10 mM). The suspension was incubated for 15 min to completely reduce the Ag^+^ ions to Ag^0^. By repeating the reduction steps, the final AgNO_3_ concentration of 300 µM was controlled. The SiO_2_@Au@Ag NPs were obtained by the centrifugation of the suspension at 8500 rpm for 15 min, then the NPs were washed thoroughly using EtOH to remove any excess reagent. The SiO_2_@Au@Ag NPs were then re-dispersed in 1 mL of absolute EtOH to obtain a 200 µg/mL SiO_2_@Au@Ag NP suspension.

### 2.3. Adsorption of 4-MPBA on the Surface of SiO_2_@Au@Ag

To adsorb 4-MPBA on the surface of SiO_2_@Au@Ag NPs, 500 µL of 2 mM 4-MPBA solution in EtOH was incubated with 500 µL EtOH contained 20 µg SiO_2_@Au@Ag NP for 1 h at 25 °C, followed by centrifugation for 15 min at 15,000 rpm to obtain the NP suspension. The prepared NPs were washed thoroughly using PBS containing 0.1% Tween 20 (PBST) to remove any excess reagent. The SiO_2_@Au@Ag@4-MPBA was then re-dispersed in 100 µL of EtOH, to obtain a 200 µg/mL SiO_2_@Au@Ag@4-MPBA suspension.

### 2.4. Behavior of SiO_2_@Au@Ag@4-MPBA in the Presence of Hydrogen Peroxide

100 µL of PBST containing 20 µg SiO_2_@Au@Ag@4-MPBA was added incubated with 100 µL of H_2_O_2_ at various concentration for 1 h at 25 °C, followed by centrifugation for 15 min at 15,000 rpm to obtain the NP suspension. The prepared NPs were washed thoroughly using PBST to remove any excess reagent. The prepared NPs was then re-dispersed in 100 µL of PBST, to obtain a 200 µg/mL NPs suspension.

### 2.5. Glucose Detection of SiO_2_@Au@Ag@4-MPBA

One hundred microliters of PBST containing 20 µg SiO_2_@Au@Ag@4-MPBA was added, incubated with 100 µL PBST that contained various concentrations of glucose, and 100 µL of GOx for 1 h at 25 °C, followed by centrifugation for 15 min at 15,000 rpm to obtain the NP suspension. The prepared NPs were washed thoroughly using PBST to remove any excess reagent. The prepared NPs was then re-dispersed in 100 µL of PBST, to obtain a 200 µg/mL NPs suspension.

### 2.6. SERS Measurement

To obtain the surface-enhanced Raman spectra, a micro-Raman system with a 10-mW 532-nm diode-pumped solid-state laser excitation source and an optical microscope (BX41, Olympus, Tokyo, Japan) was utilized. The SERS signals were collected with a back-scattering geometry using an objective lens with a magnification of 10× (0.90 NA, Olympus, Tokyo, Japan). The selected sites were measured randomly, and all the SERS spectra were integrated over a period of 5 s. The size of the laser beam spot was approximately 2.1 μm, and the SERS spectrum was obtained within the wavenumber range of 300–1700 cm^−1^.

## 3. Results and Discussion

In this study, SiO_2_@Au@Ag was first immobilized by 4-MPBA as a Raman reporter and labeling molecule. Then this suspension was added into glucose solution in the presence of the GOx enzyme. GOx enzyme catalyzed and converted glucose to gluconolactone and H_2_O_2_. The presence of H_2_O_2_ converted 4-MPBA to 4-MPheOH on the surface of SiO_2_@Au@Au. The variation of SERS signal of 4-MPBA was observed and extrapolated to the concentration of glucose in solution (Figure 1). To prepare SiO_2_@Au@Ag, silica NPs (~150 nm) were first functionalized by 3-aminopropyltriethoxysilane (APTS) to prepare aminated silica NPs. Simultaneously, colloidal Au NPs (3 nm) were synthesized by THPC and incubated with the aminated silica NPs by to prepare Au NPs seed embedded with SiO_2_ (SiO_2_@Au seed NPs), according to the method reported by Pham et al. [37,38,42,43,44]. Subsequently, the Ag NPs on the surface of SiO_2_@Au seed were grown by reducing a silver precursor (AgNO_3_) in the presence of ascorbic acid (AA) and polyvinylpyrrolidone (PVP) [37]. The silver ions reduced by AA were selectively grown onto SiO_2_@Au seed to generate the SiO_2_@Au@Ag nanostructure. According to a previous study, 4-MPBA was used for specific binding of glucose with the boronic acid motif in 4-MPBA, leading to a significant increase in the absolute intensity of the SERS signal of 4-MPBA. So, we modified the surface of SiO_2_@Au@Ag by 4-MBPA to generate SiO_2_@Au@Au@4-MPBA as a specific ligand for glucose. Subsequently, glucose was added in the reaction in the presence of glucose oxidase (GOx). GO catalyzed and converted glucose to glucolactone and hydrogen peroxide (H_2_O_2_) in the reaction. H_2_O_2_ converts 4-MPBA to 4-mercaptophenol (4-MPhOH).

The characteristics of the SiO_2_@Au@Ag NPs are first shown in Appendix A. The transmission electron microscopy (TEM) images of the SiO_2_@Au and SiO_2_@Au@Ag NPs are shown in Appendix A. Small-sized Au NPs (3 nm) were immobilized onto the surface of the SiO_2_ NPs when the colloidal Au NPs were incubated with the amine-functionalized silica NPs for 12 h at 25 °C (Appendix A). After the reduction of AgNO_3_ in the presence of PVP, the surface of the SiO_2_@Au NPs contained many Ag NPs (Appendix A). The UV–Vis spectra of SiO_2_@Au and SiO_2_@Au@Ag were investigated, as shown in Appendix A. The suspension of the SiO_2_@Au NPs exhibited the maximum peak at ~500–520 nm (Appendix A). A broad band was observed from 320–700 nm, with the maximum peak at ~450 nm, for the suspension of SiO_2_@Au@Ag NPs (Appendix A). This indicated the generation of Ag shells, in addition to the creation of hot-spot structures, on the surfaces of the SiO_2_@Au NPs, thus yielding a continuous spectrum of resonant multi-modes [37,38,42,43].

### 3.1. Adsorption of 4-MPBA on the Surface of SiO_2_@Au@Ag

It is believed that 4-MBPA can selectively interact with glucose in solution, so the SERS-active SiO_2_@Au@Ag were incubated with 4-MPBA solution for 1 h at 25 °C. Figure 2a showed the SERS bands of SiO_2_@Au@Ag@4-MPBA in ethanol solution. Compared to the SiO_2_@Au@Au without 4-MPBA (0 µM) and with 100 µM 4-MPBA, we found that the SiO_2_@Au@Ag in EtOH solution showed SERS bands at 432, 883, 1051, 1097, 1276, and 1455 cm^−1^, assigned to the SERS bands of the EtOH solution. At 100 µM 4-MPBA, clear and new bands were observed at 417, 473, 1077, 1170, 1484, and 1583 cm^−1^, indicating that 4-MPBA was successfully immobilized on the surface of SiO_2_@Au@Ag [18]. The detailed SERS bands of SiO_2_@Au@Ag@4-MPBA can be seen in Appendix A. The SERS intensity of SiO_2_@Au@Ag@4-MBPA was also examined in the presence of various concentrations of 4-MPBA from 0.1 µM to 100 µM in EtOH solution. The SERS signal of 4-MBPA increased from 0.1 to 10 µM and achieved a saturation at 50 µM. In particular, the intensities of the Raman bands at 1583 cm^−1^ increased dramatically with 4-MPBA concentration (Figure 2b). Therefore, 50 µM 4-MBPA was chosen for further study.

However, the SERS signal of EtOH was too strong compared to the SERS signal of SiO_2_@Au@Ag@4-MPBA, which can seriously affect the accuracy of analytical result. Therefore, we re-dispersed the SiO_2_@Au@Ag@4-MPBA in the PBST pH 7.0 (Figure 3a). Indeed, the SERS bands of 4-MPBA in PBST was clearly observed, compared to those in EtOH. Figure 3a showed that the typical SERS bands of 4-MBPA were dominated by bands at 417 cm^−1^, 473 cm^−1^, 824 cm^−1^, 998 cm^−1^, 1021 cm^−1^, 1077 cm^−1^, 1170 cm^−1^, 1484 cm^−1^, and 1583 cm^−1^. The SERS spectra at 998 cm^−1^, 1021 cm^−1^, and 1077 cm^−1^, which are assigned to the C-C in-plane breathing, C-H in-plane breathing, and C-C in-plane breathing and C-S stretching, respectively (Figure 3a, Appendix A). The pair of intense bands at 1586/1573 cm^−1^ is ascribed to the original and OH^-^-associated forms of C-C stretching. The SERS spectra at 824 and 417 cm^−1^ were assigned as the C-C out-plane bending and the C-S stretching with weak intensities [18,34,45,46].

### 3.2. Behavior of SiO_2_@Au@Ag@4-MPBA in the Presence of Hydrogen Peroxide

First, the SiO_2_@Au@Ag@4-MPBA were used as a SERS substrate for glucose detection in the range of 1 to 10 mM (Appendix A). According to the previous report, the boronic acid group of 4-MPBA specifically binds to glucose, leading to a significant increase in the absolute intensity of the SERS signal of 4-MPBA, which was ascribed to the orientation change and the charge transfer effect [17,45]. However, the SERS signal of SiO_2_@Au@Ag@4-MPBA at pH 7.0 in our study did not show any significantly difference in the range of 1 to 10 mM glucose. Therefore, glucose oxidase enzyme was added to the glucose solution to convert glucose to gluconolactone and hydrogen peroxide to SiO_2_@Au@Ag@4-MPBA in our study. The behavior of SiO_2_@Au@Ag@4-MPBA in PBST pH 7.0 containing 1 mg/mL H_2_O_2_ was investigated and is shown in Figure 3a.

As mentioned above, we chose the SERS band of 4-MPBA at 417, 1077, and 1583 cm^−1^ to investigate the variation of SERS band in the presence of H_2_O_2_. When SiO_2_@Au@Ag@4-MPBA was incubated with H_2_O_2_, the new bands could be observed clearly at 390 cm^−1^, 1170 cm^−1^, and 1597 cm^−1^. According to previous report, boronated molecules reacts selectively with H_2_O_2_ to convert to its corresponding phenol form, so 4-MPBA on the surface of SiO_2_@Au@Ag can converted to 4-MPheOH or thiophenol [34,46,47]. Thus, we observed the SERS band of both 4-MPheOH and thiophenol in ethanol to compare with 4-MPBA in the presence of H_2_O_2_ (Appendix A). The SERS bands of thiophenol was different from those of 4-PheOH, and 4-MPBA in the presence of H_2_O_2_ with dominant bands at 416 cm^−1^, 996 cm^−1^, 1018 cm^−1^, 1069 cm^−1^, and 1570 cm^−1^. In contrast, the SERS bands of 4-MPBA in the presence of H_2_O_2_ was similar to those of 4-MPheOH with the typical bands at 390 cm^−1^, 1170 cm^−1^, and the intense band at 1583/1597 cm^−1^ (Figure 3a). Based on these results, we concluded that 4-MPBA on the surface of SiO_2_@Au@Ag was successfully converted to 4-MPheOH in PBST pH 7.0. The effect of H_2_O_2_ concentration in the range of 10 pg/mL to 1 mg/mL on the SERS signal of 4-MPBA was also observed in Figure 3b. The bands at 390 cm^−1^, 1170 cm^−1^, and 1597 cm^−1^ increased while the bands at 417 cm^−1^ and 1583 cm^−1^ decreased dramatically with the H_2_O_2_ concentration. The SERS intensity at 1077 cm^−1^ was slightly changed. We used the SERS bands of 4-MPBA at 417 cm^−1^, 1077 cm^−1^, and 1583 cm^−1^ as intrinsic internal standards to robust quantitative detection of H_2_O_2_ in our study. The results are shown in Figure 3c and Appendix A. Indeed, the increase of H_2_O_2_ concentration led to increases of SERS ratio of I_390/417_, I_390/1077_, I_390/1583_, I_1170/417_, I_1170/1077_, I_1170/1583_, I_1597/417_, I_1597/1077_, and I_1597/1583_ (Appendix A). The dramatic increase in SERS ratio was observed in the range of 1 µg/mL to 1000 µg/mL of H_2_O_2_. The detection limit of H_2_O_2_ based on the SERS ration of I_1583/1597_ was calculated to be 0.31 µg/mL (3σ/k, where σ is the standard deviation of the blank and k is the slope of the calibration slope).

Next, the effect of experimental conditions for glucose detection on the SERS signal of SiO_2_@Au@Ag@4-MPBA in the presence of GOx were examined and optimized in Figure 4 and Appendix A.

To reduce the detection time of glucose, we attempted to combine the enzyme reaction and conversion of 4-MPBA to 4-MPheOH into one step, which is termed the “all-in-one” technique. We also conducted the glucose detection in two separate steps: The enzyme reaction followed by the conversion of 4-MPBA to 4-MPheOH, which is termed the “step-by-step” technique. For both techniques, the glucose concentration was fixed at 5 mM using 20 µg SiO_2_@Au@Ag@4-MPBA for 30 min while the GOx concentration was varied in the range of 10^−2^–10^3^ µg/mL. The results showed that the SERS signal of the SiO_2_@Au@Ag@4-MPBA at 5 mM glucose and various concentration of GOx in both the “step-by-step” and “all-in-one” techniques were quite similar (Appendix A). However, the SERS signal of SiO_2_@Au@Ag@4-MPBA in the “step-by-step” reached saturation earlier than that in the “all-in-one”. As a result, its dynamic linear range was narrower than that of “all-in-one”. Figure 4a showed that in general the SERS signal at I_390/1077_, and I_390/1583_ increased with the GOx concentration in the range of 0.1 to 100 µg/mL and achieved the saturation at 100 µg/mL GOx, while the SERS ratio at I_390/417_ increased with the GOx concentration in the range of 1 to 1000 µg/mL. Thus, 100 µg/mL GOx was utilized for further study.

The incubation time of glucose detection by the SiO_2_@Au@Ag@4-MPBA was also performed in Figure 4b and Appendix A. The SERS ratio at I_390/1077_ was saturated at 30 min, and another SERS ratio of the SiO_2_@Au@Ag@4-MPBA increased with incubation time until 1h. The gradual increase in the SERS signal of SiO_2_@Au@Ag@4-MPBA in the presence of GOx indicated that the enzyme reaction and conversion of 4-MPBA to 4-MPheOH took place simultaneously for 1h.

In addition, the effect of SiO_2_@Au@Ag@4-MPBA core-shell amount was observed in the range of 10 to 50 µg. The results can be seen in Figure 4c and Appendix A. It is well known that SERS signal depends on the Raman reporter of nanomaterials [39]. Therefore, when a greater amount of SiO_2_@Au@Ag@4-MPBA was added, more 4-MPBA were available on the surface of SiO_2_@Au@Ag, generating numerous detection sites for glucose. However, the SERS signals of the SiO_2_@Au@Ag@4-MPBA at various core-shell quantities were insignificantly different in our study, indicating that quantities of 4-MPBA molecules of SiO_2_@Au@Ag at 10 µg were enough to react with all H_2_O_2_ generated by 5 mM glucose in the enzyme reaction. However, to ensure that the SiO_2_@Au@Ag@4-MPBA is sufficient to convert H_2_O_2_ generated by higher glucose concentration in diabetes, we decided to use 20 µg SiO_2_@Au@Ag@4-MPBA for further study.

For Raman measurement, the concentration of SERS substrate is one of the most important factors affecting the SERS signal [39,41,44]. Figure 4d and Appendix A showed the effect of SiO_2_@Au@Ag@4-MPBA concentration after glucose incubation in the presence of GOx on the SERS signal of 4-MPBA. In the absence of glucose, the SERS signal of SiO_2_@Au@Ag@4-MPBA suspension significantly decreased when SiO_2_@Au@Ag@4-MPBA concentration decreased from 200 µg/mL to 100 µg/mL and the SERS signals were almost unchanged after 100 µg/mL SiO_2_@Au@Ag@4-MPBA. Meanwhile, the SERS signals of SiO_2_@Au@Ag@4-MPBA in the presence of 5 mM glucose increased slightly when SiO_2_@Au@Ag@4-MPBA decreased to 67 µg/mL. After subtracting background signal, the SERS signal achieved the highest value at 67 µg/mL. For concentrations lower than 50 µg/mL, the SERS signal of the glucose-incubated SiO_2_@Au@Ag@4-MPBA suspension decreased sharply owing to the low diffraction of the suspension.

Based on the above-mentioned results, the highest signal of glucose detection by the SiO_2_@Au@Ag@4-MPBA in the presence of GOx was achieved at 100 µg/mL GOx with 1-h incubation time using 20 µg SiO_2_@Au@Ag@4-MPBA and Raman measurement at 67 µg/mL SiO_2_@Au@Ag@4-MPBA.

After optimizing the detection conditions, the SERS spectra of the SiO_2_@Au@Ag@4-MPBA suspensions with various glucose concentrations were obtained. Figure 5 described the changes in SERS signal of SiO_2_@Au@Ag@4-MPBA after reacting with different concentrations of in glucose the range of 0.5 to 8.0 mM with and without 100 µg/mL GOx. The SERS signals of the nanomaterial suspensions at I_390/417_, I_390/1077_, I_390/1583_, I_1170/417,_ I_1170/1077, I1170/1583_, I_1597/417_, I_1597/1077_, and I_1597/1583_ increased sharply when the glucose concentration was increased from 0.5 mM to 8.0 mM (Appendix A). This implied that H_2_O_2_ was produced from GOx in the suspensions by the GOx, and that H_2_O_2_ was able to convert 4-MPBA to 4-MPheOH on the surface of SiO_2_@Au@Ag. When the concentration of glucose increased higher than 6.0 mM, the SERS peak reached saturation. This result was due to complete conversion of 4-MPBA to 4-MPheOH on the surface of SiO_2_@Au@Ag@4-MPBA. Meanwhile, the SERS signal of SiO_2_@Au@Ag@4-MPBA in glucose without GOx remained unchanged, indicating that the increase of glucose in the solution did not significantly affect the SERS of 4-MPBA on the surface of SiO_2_@Au@Ag as mentioned in Figure 2. Therefore, we concluded that the SERS signal of SiO_2_@Au@Ag@4-MPBA in the glucose solution in the presence of GOx was the result of the combination of the GOx reaction and the conversion of 4-MPBA to 4-MPheOH on the surface of SiO_2_@Au@Ag.

A linear curve-fitting procedure was utilized for calibration. A significant relationship between the SERS signal ratio and the glucose concentration was found in the experimental data points ranging from 1.0 to 8.0 mM (calibration curve: y = 0.159 x + 0.707, where x is glucose concentration, y is the SERS ration, and R^2^ = 0.99). The theoretical LOD was 0.15 mM, estimated by the 3sblank criterion. Our result demonstrates that our material can be utilized for glucose detection for diagnosis.

The interference behavior is an important factor for glucose detection since easily oxidizable species, such as ascorbic acid (AA), uric acid (UA), and bovine serum albumin (BSA), at various concentrations and fructose co-exist with glucose in blood samples [48,49,50]. Figure 5b depicted the evaluation of the selectivity of the SiO_2_@Au@Ag@4-MPBA as a SERS substrate for glucose detection at I_390/417_ in the presence of interfering species including 50 µM AA, 40 µM UA, 0.5% BSA, 1% BSA, 2% BSA, and 5 µM fructose were evaluated. Additionally, the SERS ration response of the SiO_2_@Au@Ag@4-MPBA in 5 mM glucose was examined as a reference, and the response SERS ratios at I_390/417_, I_390/1077_, I_390/1583_, I_1170/417_, I_1170/1077_, I_1170/1583_, I_1597/417_, I_1597/1077_, and I_1597/1583_ in 5 mM glucose in the presence of interferences were simultaneously observed in Appendix A. In the presence of 50 µM AA, the SERS ratio of SiO_2_@Au@Ag@4-MPBA at I_390/417_ decreased 2% to 98%, while in the presence of 40 µM UA, it decreased to 95% compared to the SERS ratio of 5 mM glucose. The presence of 0.5% BSA caused an insignificant decrease in the SERS ratio at I_390/417_ to 96%. However the SERS ratio at I_390/417_ in the presence of 1% BSA or 2% BSA showed a significant decrease to 86 and 82%, respectively, because of the adsorption of BSA on the surface of SiO_2_@Au@Ag@4-MPBA [51]. The SERS ratio at I_390/417_ of 5 mM glucose in the presence of 5 µM fructose decreased slightly to 98%. It meant that the detection of glucose by SiO_2_@Au@Ag@4-MPBA was highly selective. Thereby, the combination of GOx and SiO_2_@Au@Ag@4-MPBA exhibited excellent selectivity for glucose detection in the presence of interfering species, 50 µM AA, 40 µM UA, 0.5% BSA, and 5 µM fructose with negligible interference to the SERS signal of glucose.

The long-term storage of the SiO_2_@Au@Ag@4-MPBA was examined in Appendix A. First, 200 µ/mL SiO_2_@Au@Ag@4-MPBA was synthesized, re-dispersed in EtOH, and stored at 4 °C for one week. The SERS signal of 4-MPBA was measured every day and the SERS signal at 1583 cm^−1^ was monitored. As showed in Appendix A, the SERS signal was not decreased until seven days, indicating that the surface of SiO_2_@Au@Ag was not oxidized during storage in EtOH at 4 °C. However, the SERS signal slightly increased after four days, which might be from the partly aggregation of SiO_2_@Au@Ag.

## 4. Conclusions

We have developed a new SERS-based boronated nanoprobe of the SiO_2_@Au@Ag@4-MPBA for quantitative, selective detection of glucose in neutral condition. This work confirmed that 4-MPBA was converted to 4-MPhOH in the presence of H_2_O_2_. Moreover, it provided a new calibration curve to evaluate H_2_O_2_ species in the range of 1.0 to 1000 µg/mL with LOD as low as 0.3 µg/mL. Moreover, the SiO_2_@Au@Ag@4-MPBA for glucose detection in the presence of GOx were optimized at 100 µg/mL GOx, 100 µg/mL GOx with 1-h incubation time using 20 µg/mL SiO_2_@Au@Ag@4-MPBA and measuring Raman signal at 67 µg/mL SiO_2_@Au@Ag. At the optimized condition, the calibration curve for selective glucose detection in the range of 0.5 to 8.0 mM was successfully developed with an LOD of 0.15 mM. The combination of GOx and our nanostructure also illustrated that our SERS probe can be coupled with other enzymes to greatly expand its applicability to biologically active targets.

## Figures and Tables

**Figure 1 nanomaterials-11-00948-f001:**
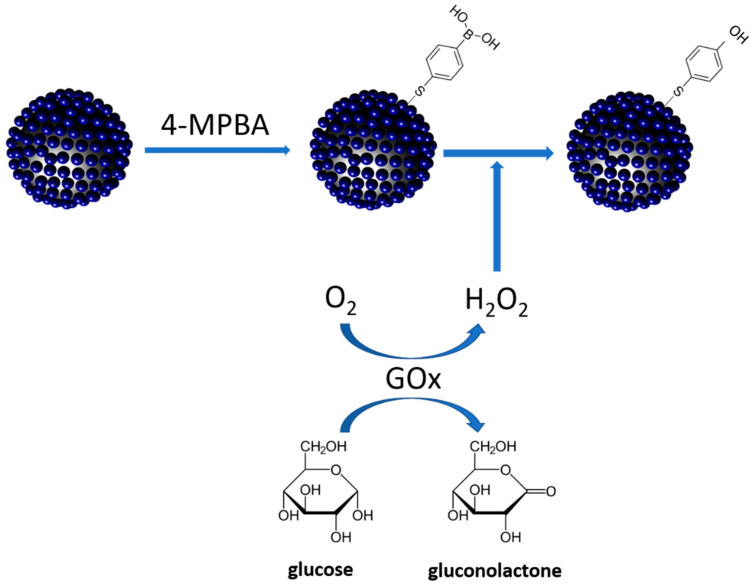
Typical illustration of glucose detection of 4-mercaptophenylboronic acid immobilized-gold-silver core-shell assembled silica nanostructure in the presence of glucose oxidase.

**Figure 2 nanomaterials-11-00948-f002:**
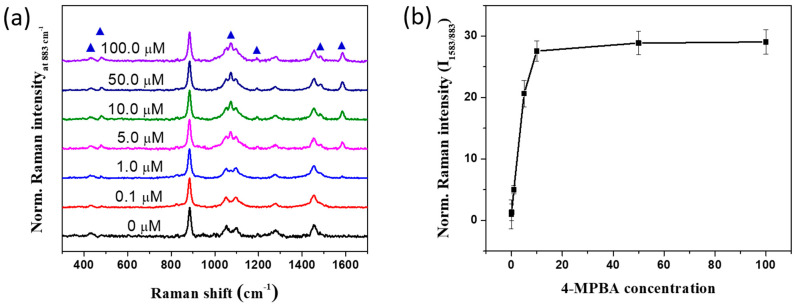
(**a**) SERS spectra and (**b**) normalized SERS intensity plot at 883 cm^−1^ of SiO_2_@Au@Ag@4-MPBA at various concentration from 0.1 to 100 µM in ethanol solution.

**Figure 3 nanomaterials-11-00948-f003:**
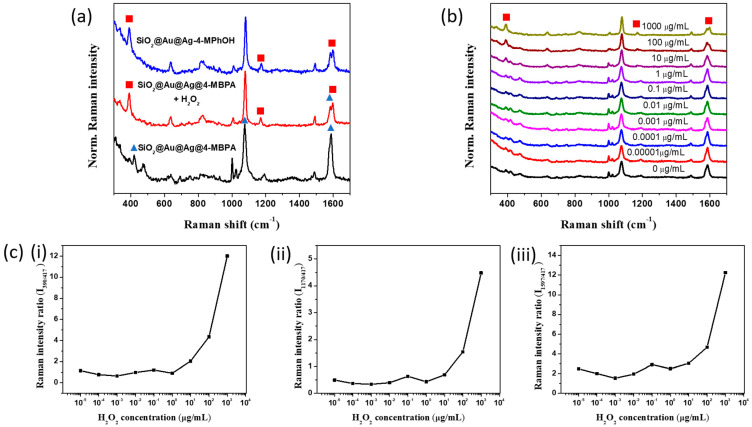
(**a**) SERS spectra and (**b**) SERS intensity of SiO_2_@Au@Ag@4-MPBA, and (**c**) SERS intensity ratio of I_390/417_, I_1170/1077_, I_1597/1583_ toward various concentration of H_2_O_2_ from 10 pg/mL to 1 mg/mL in PBST pH 7.0.

**Figure 4 nanomaterials-11-00948-f004:**
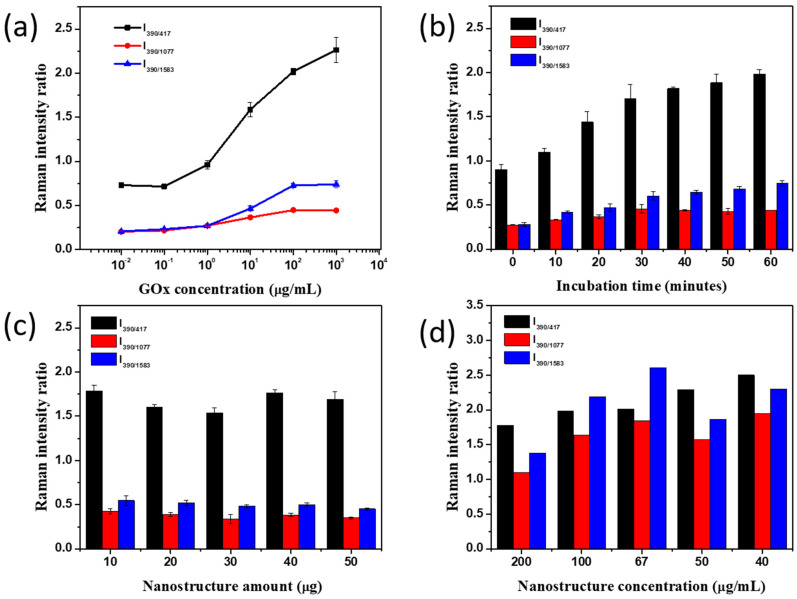
Optimization of glucose detection by SiO_2_@Au@Au@4-MPBA in all-in-one technique, (**a**) glucose oxidase concentration, (**b**) incubation time, (**c**) SiO_2_@Au@Ag@4-MBA amount, and (**d**) dilution of SiO_2_@Au@Ag for Raman measurement in PBST at I_390/417_, I_390/1077_, I_390/1583_.

**Figure 5 nanomaterials-11-00948-f005:**
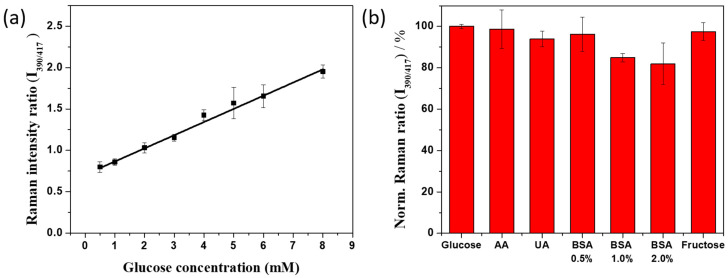
(**a**) SERS signal plot of SiO_2_@Au@Au@4-MPBA at various glucose concentrations and (**b**) the effect of interferences on the SERS signal of SiO_2_@Au@Ag@4-MPBA in 5 mM glucose and 50 µM AA, 40 µM UA, 0.5% BSA, 1% BSA, 2% BSA, and 5 µM fructose by SiO_2_@Au@Au@4-MPBA in PBST at SERS ratio of 390 and 417. The optimized condition is 20 µg SiO_2_@Au@Ag, 100 µg/mL glucose oxidase concentration for 1h, and Raman measurement at 67 µg/mL SiO_2_@Au@Ag.

## Data Availability

Data of the study is included in the main text and/or the Appendix A.

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
