# Peer review of "Glucose Detection of 4-Mercaptophenylboronic Acid-Immobilized Gold-Silver Core-Shell Assembled Silica Nanostructure by Surface Enhanced Raman Scattering"

_nanomaterials, 2021, doi:10.3390/nano11040948_

Round 1
Reviewer 1 Report
In this manuscript, SiO2@Au@Ag NPs@4-MPBA was prepared to SERS detect glucose in the presence of GOx based on the generated H2O2 converting 4-MPBA to 4-MpheOH. However, the test time is more than 1 hour, resulting in the less application cases, compared with the Gox-based electrochemical method (Glucometer).
In addition
- The Raman bands of MPBA and 4-MpheOH should be carefully assigned to understand the quantitative detection mechanism.
- The activity of GOx is dependent on the production batches, which affecting the glucose test results and the calibration method should be suggested.
- Shelf-time of SiO2@Au@Ag NPs@4-MPBA should be checked.
- The yield of H2O2 converting 4-MPBA to 4-MpheOH should be checked, which is an important fact for trace test.
- The interference from fructose should be checked.
- The ascorbic acid is a strong reducing reagent but no reaction between H2O2 and ascorbic acid is observed. The reasonable comment should be given.
- The comparison with other optical method of sensing glucose should be done.
- The blood tests by using such SERS protocol and standard method should be done to show the robustness of method.
- The abbreviation at first place in the main context should state the full name.
Author Response
Dear Reviewers:
Thank you for considering the enclosed manuscript “Glucose detection of 4-mercaptophenylboronic acid-immobilized gold-silver alloy assembled silica nanostructure by Surface enhanced Raman scattering.” (nanomaterials-1144865) for publication in the Nanomaterials as a research article.
We appreciate the comments from the reviewers who spent invaluable time and effort. We have incorporated additional modifications based on the reviewers’ thoughtful comments, which have helped us to improve the manuscript. The detailed responses to the reviewers’ comments are provided at the end of this letter.

Reviewer 2 Report
Xuan-Hung Pham and coauthors present Ag/Au-coated silica particles for the detection of glucose by surface-enhanced Raman scattering (SERS) spectroscopy. The glucose detection was enabled by the H2O2-mediated chemical conversion of 4MPBA ligands to 4MPheOH ligands. The silica particles themselves serve as substrate and carrier material for the overgrowth of the gold seeds with silver. The detection of glucose is thus achieved indirectly via the formation of 4MPheOH and allows excellent sensitivity. The results are presented comprehensibly and the manuscript possesses sufficient novelty to justify publication. The manuscript fits excellently into the subject area of the journal Nanomaterials. I am in favor of publication after minor revisions.
Comments/questions:
- The reduction of Ag was mediated using PVP. This might indicate that there could be residual coating of PVP present on the Au@Ag NPs. Is this the case and could the authors observe any SERS signals that might support this assumption? Could a residual amount of PVP interfere with the 4MPBA functionalization?
- The authors claim that BSA did not interfere with the proposed application. However, BSA is known to have a high tendency to adsorb to gold and silver surfaces (ACS Appl. Mater. Interfaces 2020, 12, 57302). As such, the authors should discuss the aspect of BSA forming a corona on the plasmonic moieties and how this might affect the colloidal stability. BSA, as well, might hinder the adsorption of molecules from the solution or interfere with the H2O2 migration to the Au@Ag surface. It might be interesting to discuss whether Raman signals of BSA are visible in the SERS spectra and if not, why.
- Line 189, typo “MBA” should be “MPBA”.
- Figure 3b: The top spectrum in yellow is hardly visible in the highlighted areas.
- Figure 5a: It would be preferable to start the y axis from 0.
Author Response

(The authors gave the same response as above.)

Reviewer 3 Report
In this manuscript, the authors measured SERS intensity of 4-mercaptophenol (4-MPhOH) adsorbed on Au/Ag core/shell nanostructures on SiO2 particle to detect amount of glucose. Fabrication of the nanostructures can be controlled on SiO2. The 4-MPhOH was made from 4-mercaptophenyl boronic acid (4-MPBA) adsorbed on the nanoparticles by H2O2, which was made by glucose oxidase. The optimum conditions for the detection of glucose were comprehensively investigated, and then the calibration curve from 1.0 to 8.0 mM was developed. The curve was not interfered by the coexistence. Consequently, the method is indirect detection via the enzyme reaction following by the conversion by H2O2, but it is still an interesting method. This manuscript may be publishable after the major revision as follows.
Line 18: The nanoparticles consist on Au core and Ag shell, as written in Line 62, rather than the alloy. In the text, "alloy" should be wholly replaced with "core/shell nanostructure" and so on. Moreover, the "SiO2@Au@Ag@4-MPBA" can be misleading expression; the SiO2, Au, and Ag seem to be core, inner, and outer shell, respectively, in an opposite way of shelled-isolated nanoparticle-enhanced Raman spectroscopy (SHINERS).
Line 41: The paper of Anal. Chem. 2010, 82, 1342-1348 can be cited here, because HbA1c has been detected by SERS.
Lines 160-165: Figure S1a should be displayed in not SI but the main text, because information about the size, shape, and arrangement of the nanoparticles will attract attention.
Lines 174-175: It is written as "Figure 2 and Figure 3a showed that the typical SERS bands of 4-MPBA..." But, these spectra do not seem to originate from the same species; in the latter, the peak at 883 cm-1 disappears, and the peak at ~1600 cm-1 is much stronger than that at ~1475 cm-1 (neighbor to the left of the yellow area in Figure 2a) unlike the former. In the peak assignments, the wavenumbers seem to be barely consistent with the spectra in Figure 2a and 3a, and thus the corresponding wavenumber should be written in the spectra. Moreover, also the peaks of 4-MPhOH should be assigned to (the same mode as 4-MPBA?) and can be expressed in Table to compare with those of 4-MPBA.
Figure 3: Nine ratios can be derived from the combination of 3 peaks of 4-MPBA with 3 new bands, as written in Lines 201-203. The authors do not have to show all of the results in the main text, but can show those in Supplementary. For the ratio at 1077 cm-1 to 1583 cm-1, neither of the peaks are the new bands after the incubation with H2O2.
Additionally, there are minor problems as follows.
Line 20 and 23: The formal name of "4-MPhOH" and "GOx" should be written for the first time, respectively.
Line 111 etc.: What is "PBST"? Is it PBS? If so, all of "PBST" in the text should be revised.
Figure 2b: The unit of X-axis is missing.
Figure 4a: Which result does the figure show, by the all-in-one or step-by-step method?
Line 256: Figure 4a -> Figure 4c
Line 284: It is written as "20 µg/mL". But, this may be not concentration but amount of the particles, and thus "/mL" should be omitted.
Line 285: In "SiO2@Au@Ag" (this can be misleading expression as above-mentioned), 4-MPBA is missing.
Lines 290-293: Which figure is explained by these sentences? The figure number should be written.
Author Response

(The authors gave the same response as above.)

Round 2
Reviewer 1 Report
n revision, authors partly responded my concerns but the test for blood samples is still absent.
(1) For application, the shelf-time stability is quite crucial and in Figure S10, with the time undergoing up to 7 days, the intensity elevates about 15% compared to one that is freshly prepared. Please comment this issue.
(2) A paper regarding Enzyme-Free Tandem Reaction Strategy –based SERS detection of glucose (ACS Appl. Mater. Interfaces 2020, 12, 55324−55330) shows superior selectivity and selectivity but no citation and no comment in this MS.
Author Response
We appreciate the comments from the editor and reviewers who spent invaluable time and effort. We have incorporated additional modifications based on the editor and reviewers’ thoughtful comments, which have helped us to improve the manuscript.

Reviewer 3 Report
The authors have properly revised the manuscript and added the new figures into the supplementary information.
Author Response

(The authors gave the same response as above.)
